# Proposal for a New Diagnostic Histopathological Approach in the Evaluation of Ki-67 in GEP-NETs

**DOI:** 10.3390/diagnostics12081960

**Published:** 2022-08-13

**Authors:** Pinuccia Faviana, Laura Boldrini, Carlo Gentile, Paola Anna Erba, Enrico Sammarco, Francesco Bartoli, Enrica Esposito, Luca Galli, Piero Vincenzo Lippolis, Massimo Bardi

**Affiliations:** 1Department of Surgical, Medical, Molecular Pathology and Critical Area, University of Pisa, 56126 Pisa, Italy; 2Azienda Ospedaliera di Catanzaro “Pugliese Ciaccio”, 88100 Catanzaro, Italy; 3Department of Translational Research and New Technologies in Medicine and Surgery, University of Pisa, 56126 Pisa, Italy; 4General and Peritoneal Surgery, Department of Surgery, Azienda Ospedaliero-Universitaria Pisana, 56124 Pisa, Italy; 5Department of Psychology and Neuroscience, Randolph-Macon College, Ashland, VA 23005, USA

**Keywords:** neuroendocrine tumors, Ki67, immunohistochemistry, digital analysis

## Abstract

Introduction: Studies have shown that the Ki-67 index is a valuable biomarker for the diagnosis, and classification of gastro-entero-pancreatic neuroendocrine tumors (GEP-NETs). We re-evaluated the expression of Ki-67 based on the intensity of the stain, basing our hypothesis on the fact that the Ki-67 protein is continuously degraded. Background: The aim was to evaluate whether a new scoring method would be more effective in classifying NETs by reducing staining heterogeneity. Methods: Patients with GEP-NET (n = 87) were analyzed. The classification difference between the two methods was determined. Results: The classification changed significantly when the Ki-67 semiquantal index was used. The percentage of G1 patients increased from 18.4% to 60.9%, while the G2 patients decreased from 66.7% to 29.9% and the G3 patients also decreased from 14.9% to 9.2%. Moreover, it was found that the traditional Ki-67 was not significantly related to the overall survival (OS), whereas the semiquantal Ki-67 was significantly related to the OS. Conclusions: The new quantification was a better predictor of OS and of tumor classification. Therefore, it could be used both as a marker of proliferation and as a tool to map tumor dynamics that can influence the diagnosis and guide the choice of therapy.

## 1. Introduction

Currently, the classification system of NETs is based on the evaluation of proliferation with incremental intervals of mitosis and on the expression of Ki-67. Counts are generally expressed as the number of mitotic cells per mm^2^, ideally counted up to 10 mm^2^ to increase accuracy. Several studies have indicated that, due to the large differences between observers in the count, the validity and reproducibility of the Ki-67 index are clearly superior to those of the mitotic count [1]. In theory, any cancer cell with detectable Ki-67 is proliferating and therefore Ki-67 has been accepted as a very sensitive proliferation marker. Many studies over the past decade have shown that the Ki-67 index is a valuable biomarker for the diagnosis, classification, and stratification of cancer prognosis [2,3,4,5].

However, problems remain about how to accurately measure Ki-67 expression. Over the years, various methods have been developed to quantify Ki-67: manual estimation by a pathologist who counts in real time through the eyepiece of the microscope [6]; manual counting using a printed image acquired using a camera mounted on a microscope with ×20 [7] magnification; computer-assisted quantification by digital image analysis (DIA) [8]. A precise assessment of the Ki-67 proliferation index is required for a rigorous classification of NETs. Furthermore, the reproducibility of the Ki-67 evaluation can be influenced by several technical factors such as the type of sample (biopsy or fine needle aspiration cytology), the staining technique and the type of antibody used [8]. Regardless of the method, there is a lack of consistency in the evaluation of Ki-67 expression when evaluating borderline values at the threshold for a classification change, both between G1 and G2 (range 2 to 5%) and between G2 and G3 (from 15% to >20%). At first glance, DIA appears to be more reliable than “eyeballing” because it can significantly reduce intra- and inter-observer variability [9,10]. In fact, several studies have indicated that computational methods such as “eyeball” estimates turned out to be the least reliable. However, DIA has other drawbacks, such as setting subjective thresholds for positive counts and a significant increase in the cost of the test, which often means a possible reduction in its broad applicability due to a lack of specialized machinery and trained personnel.

The threshold for positive count can change due to both the heterogeneity of Ki-67 expression within a tumor and the intensity of the staining that indicates that expression. In fact, it is known that the expression of Ki-67 constantly increases during the cell proliferation cycle, reaching a peak at G2/M. Current standards suggest including any discernible staining (nuclear or diffuse, weak or strong) in the evaluation of the expression of Ki-67 [11,12], in accordance with the recommendations of the World Health Organization (WHO). As a consequence of these recommendations, the grading of the classification using Ki-67 is significantly higher than using the mitotic count. Indeed, up to one-third of tumors have higher grades based on Ki-67 than classifications based on mitotic counts [13].

Another problem could be the possible inclusion, in the Ki-67 count, of non-tumor proliferating cells residing within the tumor sample such as intratumoral endothelial cells, the underlying epithelium (e.g., glands, crypts, etc.), and lymphocytes. To minimize the number of false positives produced by the evaluation of Ki-67 expression, it is also important to distinguish negative tumor cells from stromal cells, which do not need to be counted. Stromal cells are typically smaller, spindle-shaped, and generally surround the clusters and nests of cancer cells. The inclusion of these Ki-67 positive non-neoplastic cells (false positives regarding neoplastic activity) can significantly increase the overall count and, on average, are sufficient to increase the classification of most tumors by at least one degree. Considering that, in most cases, false positives are characterized by lighter coloration, it is important to reconsider the current guidelines on the evaluation of Ki-67. Some researchers have raised the idea that a more accurate assessment of cell proliferation can be obtained by using the intensity distribution degrees in the immunohistochemical stains of Ki-67 [11], rather than simply considering the expression of Ki-67 as an “on-off” binary switch: “on” during cell proliferation is “turned off” during quiescence and senescence.

Finally, it should be considered that the empirical observation of paraffin sections subjected to immunohistochemistry (IHC) for the determination of Ki-67 shows that typical/atypical mitoses are easily recognizable. Other positive nuclei are routinely identified without considering differences in Ki-67 immunoreactivity, which are also easily distinguishable. Researchers have described signs other than Ki-67 patterns in normal and cancerous cells: spots or spots, loose or agglomerated grains, highly developed and homogeneous mitotic nuclei, and chromosomes [14,15,16,17,18,19,20,21].

These studies have shown that the position of Ki-67 differs in the different phases of the cell cycle; in fact, during the interphase, Ki-67 is involved in the organization of heterochromatin and in the nucleolar periphery [22,23,24]. Its expression is already low at the beginning of the G1 phase. These features have been observed with immunofluorescence techniques in HeLa cells and are indicative of the existence of two localization patterns of Ki-67, a homogeneous patch (EG1) and a late phase (LG1) [19,25]. In the S phase, the size of the Ki-67 points increases until it becomes dense granules that are distributed throughout the nucleus [20,22,26]. Ki-67 expression increases from the S phase onwards with a progressive increase during G2 and reaches its maximum in mitosis [27,28,29,30] (Figure 1).

In the present work, therefore, we have decided to evaluate the expression of Ki-67 in GEP-NETs not as a binary state, but on the basis of the intensity of the coloring. We based our hypothesis on the fact that the Ki-67 protein is continuously degraded. Considering these biological mechanisms, we argue that a simple Ki-67 score as positive or negative in a tumor biopsy can be oversimplification [30]. Therefore, the main objective of this study was to develop and propose a new method for evaluating Ki-67 expression by focusing on the intensity of staining provided by DIA. Since we used both computer-aided quantification and researcher assessment of the characteristics of the stained area (both intensity and classical neuroendocrine morphology), we have referred to this method as semiquantal DIA. To determine the proliferative activity of GEP-NETs, only the highly colored (near black) nuclei in the neoplastic tissue were considered actively proliferating, hence neoplastic cells in the G2/M phase of the cell cycle. We compared the tumor classification of GEP-NETs using currently used methods (binary state of Ki-67 expression) and our semiquantal DIA assessment to assess the significance of grade changes, with the aim of improving the accuracy of the current classification and therefore to offer patients suffering from GEP-NETs more accurate therapeutic solutions. Our main hypothesis was that proposed semiquantal evaluation would reduce the overall number of higher-grade classifications (G3 and G3) reflecting better the actual clinical data. Our secondary aim was to demonstrate that the new Ki-67 evaluation would be able to more accurately predict overall survival of the patients.

## 2. Materials and Methods

### 2.1. Patients and Classification

We retrospectively selected 87 patients who underwent surgical resection of gastroenteropancreatic NETs at the Unit of Surgery at Pisa University between 2012 and 2020 and satisfied the inclusion criteria. Patients were included if the biopsy contained a cell block in which the presence of NET was demonstrated by multiple immunohistochemistry indices (chromogranin, synaptophysin, and the Ki-67 index) and the estimated number of tumor cells per block was higher than 100 cells. There were 45 males and 42 females, with a mean age of 64.5 years, median age 67.2 years, range of 27–94 years.

Almost all tumor grades (from G1 to G3) were represented in our final sample. A pathologist (P.F.) formulated histologic diagnoses according to the WHO 2017 [18] and 2019 [19]: G1, well differentiated, with a mitotic rate from 0 to 1 per 10 high-power fields (HPF) or a Ki67 index from 0% to 2%; G2, well differentiated, with a mitotic rate from 2 to 20 per 10 HPF or a Ki-67 index from 3% to 20%; NET G3, well-differentiated and poorly differentiated neuroendocrine carcinomas (NEC G3) with a mitotic rate greater than 20 per 10 HPF or a Ki-67 index greater than 20%. In our sample, there were only 6 neuroendocrine carcinomas (NEC G3) and they were not used for statistical analysis. (Table 1).

### 2.2. Statement of Ethics

This study was conducted in accordance with the World Medical Association Declaration of Helsinki and approved by the Ethical Committee of the University of Pisa (Protocol # 9989 from 20 February 2019). Written informed consent was obtained from each patient.

### 2.3. Ki-67 Immunohistochemistry

Formalin-fixed, paraffin-embedded cell blocks were used for immunohistochemical staining. The protein expression of Ki-67 was assessed on FFPE tumor tissue samples using IHC. Ki-67 monoclonal antibody (rabbit monoclonal primary antibody anti-Ki-67, 30.9; Roche SpA, Monza, Italy) was used with the avidin-biotin peroxidase method for developing the immunoreaction. Immunohistochemical analysis was performed on the Ventana Medical System, with appropriate positive controls run for all cases. Staining of a known Ki-67-positive case and omitting the first antibody were used as positive and negative controls, respectively. 

The number of Ki-67 immunoreactive cells was determined by scoring a minimum of 10 high-power fields (magnification, 40×) and counting the number of immunoreactive cells (PS) out of the total epithelial cells analyzed in each field of view. The new assessment was calculated on the bases of two intensities of nuclear staining: “low”, defined as pale brown intensity, and “high”, defined as a dark brown or black. Only the percentage of nuclei stained with high intensity was used to compute the new Ki-67 assessment (see Figure 1 for representative non-pancreatic (a and b) and pancreatic (c and d) cases, respectively. The program used for the image analysis is “NIS Elements BR (Basic Research) Vers. 5.20” (Nikon) (Nikon Europe B.V., Stroombaan 14, Amstelveen, The Netherlands). The new evaluation was calculated on the basis of two intensity of nuclear staining:” low “, defined as brown intensity light, and “tall”, defined as dark brown or black, and on the size of the cell. For the count, restrictions have been inserted to specify the characteristics of the cores that act as functions to include or exclude them within the image, the selected function, for example, in our study is the “Maximum intensity” restriction to not include cores small and low-intensity staining.

### 2.4. Mitotic Count (MC)

The mitotic index was scored as the total number of mitotic figures in non-overlapping 10 consecutive HPF (original magnification ×40) where they were most prevalent on hematoxylin and eosin (H&E) sections.

### 2.5. Statistical Analysis

Correlations among variables were calculated using Pearson’s r. Differences in average counts were measured using *t*-tests. Differences in frequency of classification were measured using χ^2^ analyses. Survival analysis was performed using the Kaplan–Meyer method with the Mantel–Cox log-rank test for statistical significance. All statistics were two-tailed, and the significance level was set as α = 0.05. Statistical analyses were performed using the SPSS 27.0 software (IBM, Chicago, IL, USA).

## 3. Results

The study population included 87 patients with GEP-NETs. Most cases (55%) were pancreatic NETs; the other 39 cases (45%) were distributed from the ileum, small intestine, appendix, colon, stomach, duodenum, and liver. The average age of the patients was 64.5 ± 14.7 years (range = 27.2–94.7 years). The average time passed since the first diagnosis was 4.64 ± 1.9 years (range = 1.6–9.5 years). Our sample was evenly split between females (51.7%) and males (48.3%). The disease-free interval (DFI) ranged from 60 days to 20 years (average = 4.56 ± 2.5 years) and was not significantly correlated to the three measures of cellular proliferation (number of mitoses: *p* = 0.503; traditional Ki-67: *p* = 0.196; semiquantal Ki-67: *p* = 0.146). Most patients were still alive at the time of the data collection (91.2%), with about 70% of them who were no longer under any treatment.

The average amount of mitosis per mm^2^ was 6.82 ± 15.4 (range = 0–113). The average number of cells expressing Ki-67 varied significantly according to the method used (t_86_ = 5.46; *p* < 0.001—Figure 2): using the traditional evaluation, the number of K-67-ir cells 13.4 ± 22.3 (range = 1–90); using the semiquantal Ki-67, the average number decreased to 7.9 ± 16.2 (range = 0–70). All measures of cellular proliferation were highly correlated (Table 2, Figure 3).

Number of mitoses, traditional Ki-67, and semiquantal Ki-67 did not differ by sex (all *p*-values > 0.43), but there was a notable difference regarding the location (pancreatic vs. non-pancreatic): whereas both the mitotic count and semiquantal Ki-67 were significantly different (mitotic count: t_86_ = 4.38; *p* = 0.039; semiquantal Ki-67: t_86_ = 6.12; *p* = 0.015), the traditional evaluation of Ki-67 was not, although it was very close to reach the significance level (t_86_ = 3.66; *p* = 0.059—Figure 4). Age and time since the first diagnosis were not related to any of the three measures of cellular differentiation (Table 3).

The classification of patients changed significantly using the traditional and the semiquantal Ki-67 evaluation (χ^2^_4_ = 63.42, *p* < 0.001). Using the proposed semiquantal method, the percentage of G1 patients increased from 18.4% to 60.9%, whereas G2 patients decreased from 66.7% to 29.9% and G3 patients also decreased from 14.9% to 9.2% (Table 4).

Overall survival for the patients was entered as the time variables in several Kaplan–Mayer analyses based on the traditional Ki-67, the semiquantal Ki-67, and the mitotic count. It was found that the traditional Ki-67 was not significantly related to the OS (Mantel–Cox = 0.026, *p* = 0.872—Figure 5A), whereas both the semiquantal Ki-67 (Mantel–Cox = 4.343, *p* = 0.037—Figure 5B) and the mitotic count (Mantel–Cox = 4.343, *p* = 0.037—Figure 5C) were significantly related to the OS, indicating that the new semiquantal Ki-67 was a better predictor of overall survivability than the traditional measure.

## 4. Discussion

Ki-67 has always been considered a necessary biomarker for the identification of the percentage of cellular fraction in the cell cycle [31,32], a necessary tool for prognosis, indication and monitoring of therapy, and the identification of new pharmacological targets [33,34,35,36].

This study aimed to re-evaluate the immunopositivity of Ki-67 in GEP-NET tissue samples. The position of Ki-67 during the cell cycle in cultured cells has been shown to be very important for the characterization of the position of Ki-67 during the interphase and mitosis [17,25,37]. Transmission electron microscopy demonstrated the position of Ki-67 in the nucleolus and flow cytometry provided information on Ki-67 during the cell cycle [15,16,19,20].

Various studies have shown that immediately after the anaphase, Ki-67 continuously degrades and disappears in G0. If daughter cells remain in the cycle, the tiny residual Ki-67 can be added to G1 at the start of Ki-67 production [18,38] and so can represent weak Ki-67 immunopositivity cells in G0 that still have Ki -67 residues in the degradation process [38]. This is an important aspect as this core in G0 could be misclassified as in the early G1.

The G2 phases, on the other hand, have been described by Braun et al. [26] who showed that in G2 the expression of Ki-67 increases significantly and occupies the entire nucleus when viewed under a light microscope.

We hypothesized that this new method would reduce the number of high-grade classifications (main hypothesis) and be more representative of the actual clinical progression (secondary hypothesis). Our main results confirmed both these hypotheses: the current methods for evaluating the Ki-67 index significantly inflated the classification of NETs in our sample, increasing the classification from G1 to G2 and from G2 to G3 in more than 30% of the 87 patients. Logistic regression also found that the most accurate predictor of the patients OS was the semiquantal Ki-67 method, which was confirmed by survival analysis performed using the Kaplan–Mayer analysis.

Current standards indicate that the quantification of the Ki67 index can be considered acceptable counting between 500 and 2000 tumor cells in the area with the highest concentration of positive nuclei. A recent study [22] has shown that the size of the field can affect the quantification of Ki-67 and revealed a significant issue as well in the reliability of the number of positive cells counted among pathologists. This additional consideration supports our hypothesis that staining intensity should be considered because by focusing on high-intensity stains, it is easier to increase user-dependent reliability in tissue sample scoring. In our sample we observed that inter-observer reliability usually increases with staining intensity, thus making both the evaluation and classification more consistent. Considering the prognostic and therapeutic consequences of a correct NETs classification, we believe that even a moderate improvement in power and inter-observer reliability would be an important step in improving our ability to intervene effectively.

Ki-67 is a non-histone nuclear protein closely associated with somatic proliferation [3]. Although antibodies produced against human Ki-67 protein have paved the way for immunohistochemical evaluation of cell proliferation, Ki-67 activation is not specific for neuroendocrine cells [3,5].

Thus, it can be sensitive to the proliferation activity of cells in the tumor microenvironment, such as stromal cells, blood vessels, and inflammatory cells [13,39,40]. Consequently, including any activity in the final count will necessarily increase the total percentage of nuclei considered positive. This issue has been known for several decades. Two studies published in the 1990s showed a strong correlation between Ki-67 index and MC in breast cancer [41] and mixed cases of colorectal adenocarcinoma, mammary carcinoma, squamous cell carcinoma, non-small cell lung cancer, and small cell lung cancer [42]. In both studies, MC was performed manually, based on cellular morphology. Recently, Huang et al. [43] explored the impact of MC and Ki-67 scores in 41 cases of neuroendocrine tumors of variable histological grades; both manual and pathologist-supervised digital image analysis methods were used showing a strong correlation between quantification of mitosis and Ki-67 index only in the tumors with peak Ki-67 index less than 30%. Their work clearly highlighted that several variables such as the selection of hotspots, field size, and especially threshold could affect accurate quantification of mitosis and especially the Ki-67 index regardless of the methodology used. Moreover, they used phosphorylation of histone H3 (PHH3) as a marker for cell mitotic activity [44]; it was a time-consuming as well as a more expensive method than our practice based on morphology. Our data strongly suggested that focusing on high-intensity stains and excluding all the surrounding Ki-67 activity, we can perform a better evaluation of the actual mitotic activity. The traditional method is a very effective and proven diagnostic tool for pathologists routinely estimate cell proliferation rates, and it is generally used to effectively predict prognosis in several cancers, including neuroendocrine tumors. However, our data suggested that there are ways to improve this evaluation, such as using high-intensity stains of Ki-67. Considering that our data included a heterogeneous set of patients with a limited number of cases, our results need to be validated by further studies before reaching definitive conclusions.

In conclusion, our data showed that a semiquantitative evaluation of the Ki-67 index based on high intensity stains may offer a reliable alternative to the current standards for the classification of NETs., especially when there could be doubts about the origin of cellular proliferation. Accurate detection of NET activity early can be a quick and feasible clinical practice to improve the stratification of NETs and, therefore, the selected therapeutic approach. Further studies will be needed to assess the exact extent of the benefits using a semiquantal Ki-67 index assessment for NETs., considering that large datasets are necessary to confirm our preliminary results.

The new analysis of the intensity of immunostaining with Ki-67 has allowed us to re-quantify the positive percentages of Ki-67, which can be used not only as a marker of proliferation but also as a tool for mapping the tumor dynamics that can affect cancer diagnosis and prognosis and guide the choice of therapy.

## Figures and Tables

**Figure 1 diagnostics-12-01960-f001:**
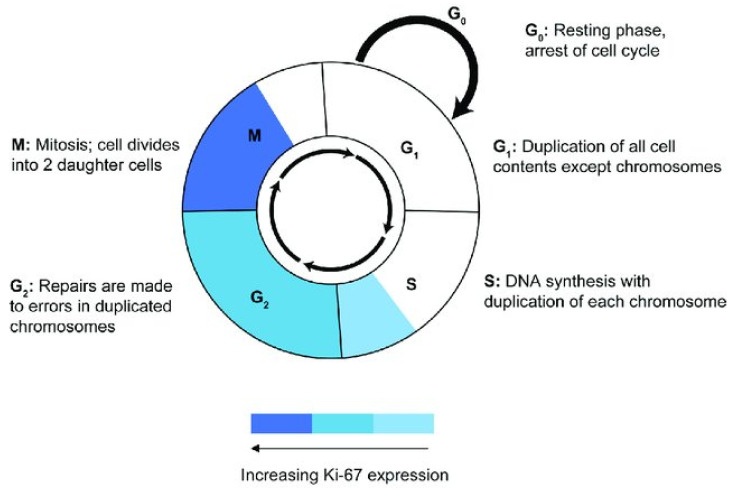
Schematic of Ki-67 expression during the cell cycle phases, peak expression and high staining intensity during the M phase [10].

**Figure 2 diagnostics-12-01960-f002:**
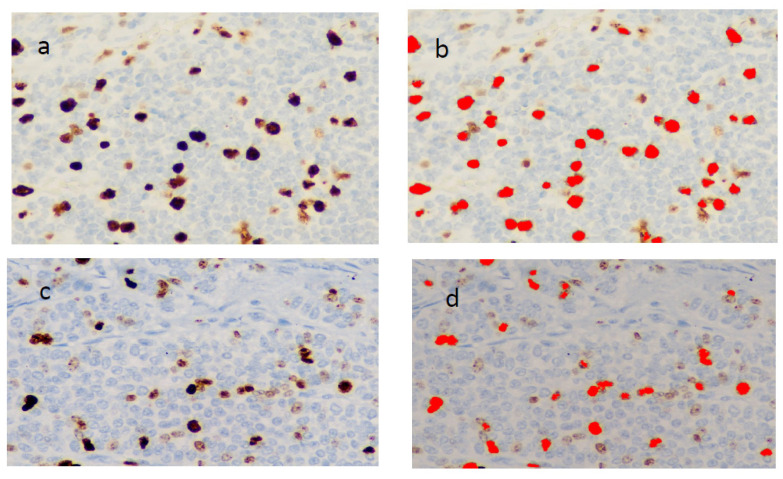
Evaluation of Ki67 proliferation index: in (**a**) and (**c**) immunostaining with Ki-67, 40× objective. In (**b**) and (**d**) the calculation was performed using specialized software, tumor nuclei considered positive for Ki-67 are indicated in red.

**Figure 3 diagnostics-12-01960-f003:**
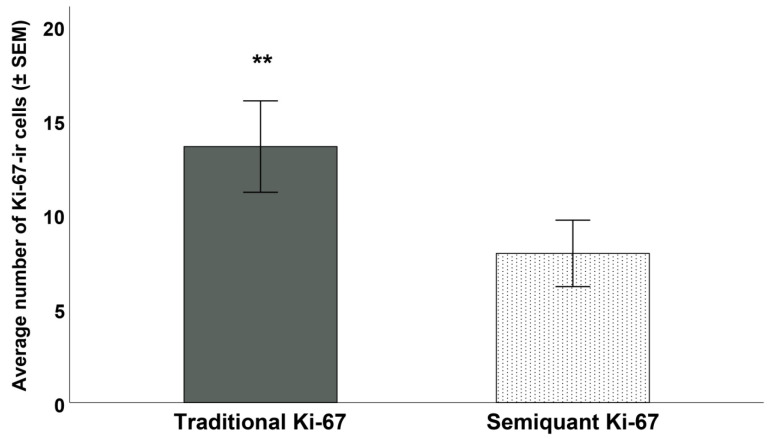
The average number of cells expressing Ki-67 varied significantly according to the method used (** = *p* < 0.001). On the left the average values counted using the traditional evaluation is shown; on the right the new semiquantal evaluation was used.

**Figure 4 diagnostics-12-01960-f004:**
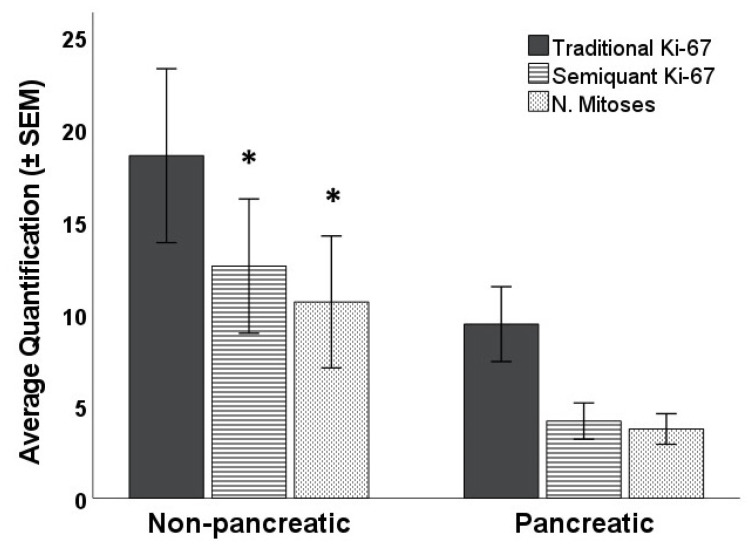
Number of mitoses, traditional Ki-67, and semiquantal Ki-67 difference by NETs location (pancreatic vs. non-pancreatic). (* = *p* < 0.05).

**Figure 5 diagnostics-12-01960-f005:**
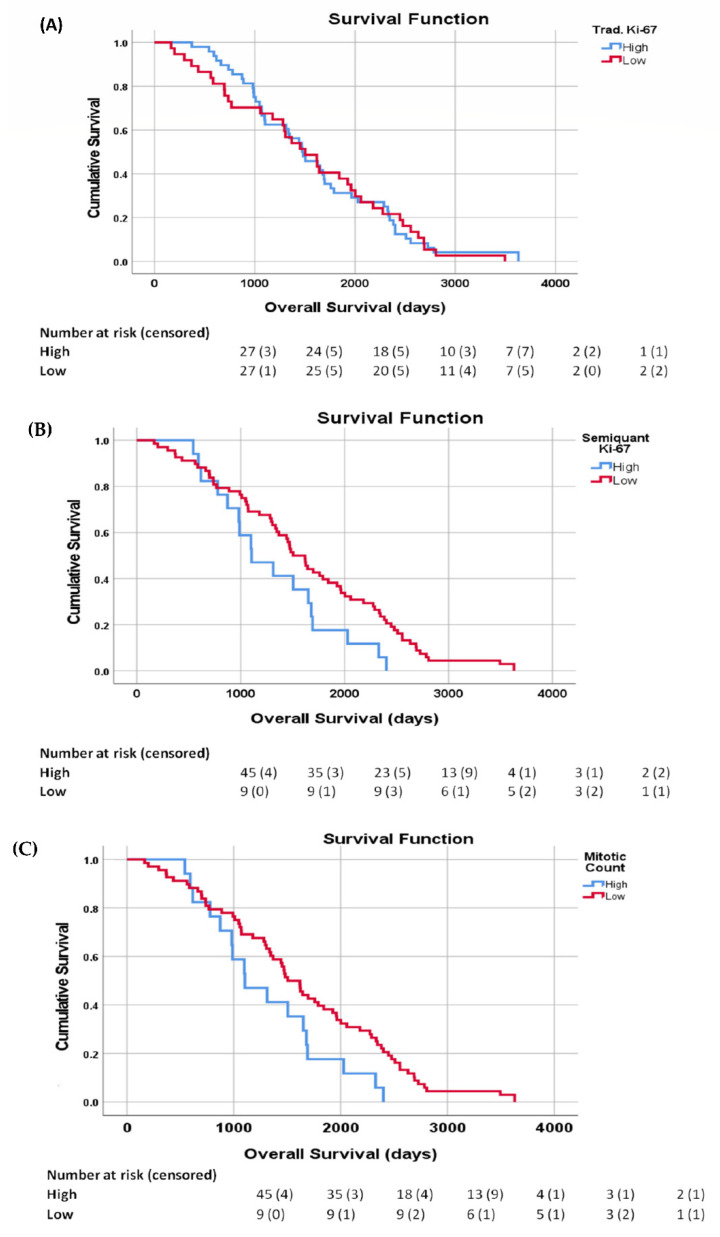
Kaplan–Meier of the overall survival (OS) by evaluation method. The number of events at risk and censored is included in the tables underneath. (**A**) Cumulative survival for high/low traditional Ki-67 evaluation; (**B**) cumulative survival for the new high/low semiquantal Ki-67 evaluation; (**C**) cumulative survival for high/low mitotic count.

**Table 1 diagnostics-12-01960-t001:** Clinical data.

	All Patients (n = 87)	G1 (N. Patients)	G2 (N. Patients)	G3 NET (N. Patients)
**Age-Median (Range)**	55 years (27–94 years)			
<49	n = 33	10	16	7
50–69	n = 32	3	23	6
>70	n = 22	3	19	0
**Sex**				
Female	n = 44	9	29	6
Male	n = 43	7	29	7
**Location**				
Pancreatic	n = 48	4	40	4
Non-Pancreatic	n = 39	12	18	9
**DFI-Median (Range)**	4.03 years (0.2–9.4 years)			

**Table 2 diagnostics-12-01960-t002:** Correlation among the measure of cellular proliferation.

		Semiquant Ki-67	n° Mitosi
Traditional Ki-67	Pearson Correlation	0.925 **	0.824 **
	Sig. (2-tailed)	<0.001	<0.001
Semiquant Ki-67	Pearson Correlation		0.703 **
	Sig. (2-tailed)		<0.001

** Correlation is significant at the 0.01 level (2-tailed).

**Table 3 diagnostics-12-01960-t003:** Correlation between clinical data and measures of cellular proliferation.

		Traditional Ki-67	Semiquant Ki-67	n° Mitosi
Age	Pearson Correlation	0.151	0.118	0.138
	Sig. (2-tailed)	0.164	0.279	0.203
Time since Diagnosis	Pearson Correlation	−0.016	−0.054	−0.001
	Sig. (2-tailed)	0.887	0.619	0.980

**Table 4 diagnostics-12-01960-t004:** Difference in the classification of patients (G1, G2, G3) using the proposed new method.

Traditional Ki-67					
		N. of Patients	Percent	Valid Percent	Cumulative Percent
Valid	G1	16	18.4	18.4	18.4
	G2	58	66.7	66.7	85.1
	G3	13	14.9	14.9	100.0
	Total	87	100.0	100.0	
**Semiquent Ki-67**					
		**N. of Patients**	**Percent**	**Valid Percent**	**Cumulative Percent**
Valid	G1	53	60.9	60.9	60.9
	G2	26	29.9	29.9	90.8
	G3	8	9.2	9.2	100.0
	Total	87	100.0	100.0	

## Data Availability

Not applicable.

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
