# Peer review of "Proposal for a New Diagnostic Histopathological Approach in the Evaluation of Ki-67 in GEP-NETs"

_diagnostics, 2022, doi:10.3390/diagnostics12081960_

Round 1

Reviewer 1 Report

I read with interest the manuscript reporting Ki67 semiquantitative assessment in predicting survival outcomes in GEPNET. Authors have included resonable hypothesis, conducted appropriate tests to validate the hypothesis and have given good introduction to support their thoughts. The experimental design is robust and results support the hypothesis. Overall, i find it difficult to comment for improvement in this manuscript! and it is appropriate in its current form. 

I have minor edits to suggest

1.  signficantlyu in line 199

2. Consider adding a graphic or cartoon on cell cycle phases and ki-67 expression to enhance reader experience of manuscript

3. Acknowledge that this results apply to heterogenous organ NETs in GI track and results need to be validated by others before these results qualify as a "game-changing" breakthrough in GEPNET oncology care. 

Author Response

  • Signficantlyu in line 199

We corrected several typos including the one above indicated.

  • Consider adding a graphic or cartoon on cell cycle phases and ki-67 expression to enhance reader experience of manuscript

We have added an example (figure 1) showing Ki-67 expression during the cell cycle phases.

  • Acknowledge that this results apply to heterogenous organ NETs in GI track and results need to be validated by others before these results qualify as a "game-changing" breakthrough in GEPNET oncology care. 

We added a sentence in the Discussion, among the other limitations, acknowledging this point.

Reviewer 2 Report

This study investigated a new evaluation method for the Ki-67 expression of GEP-NET. The authors concluded the new method allowed more accurate assessment of tumor grade and was significantly associated with OS in patients with GEP-NET. This study is a very interesting study, but there are several concerns that need to be addressed.

1) The introduction part is redundant. It would be better to be more concise.

2) It would be better to add a table showing the clinical characteristics of patients.

3) The authors described NET G3 included NEC G3 (lines 146­­­–147), but NET and NEC are different disease concepts. Clinical data and OS for NET G3 and NEC G3 need to be evaluated separately.

4) Although the authors described “whereas mitotic count and semiquant Ki-67 were significantly different, the traditional evaluation of Ki-67 was not”, in Figure 3, traditional Ki-67 is marked with an asterisk and N. mitoses in not. Which is correct?

5) I couldn't understand what the authors mean by table 2. Is this necessary information?

6) In figure 4, the cumulative survival rates reach. Is this correct results? Please add the points of censored or events and number of patients at risk in the survival curves

7) There are a lot of evidence that the traditional method is significantly related to prognosis. Please discuss why the traditional method did not correlate with OS of GEP-NET patients in this study.

Author Response

  • The introduction part is redundant. It would be better to be more concise.

We eliminated a section in the last paragraph of the Introduction and reduced the amount of information provided about the proliferation phases in the 6th paragraph of the Introduction to make it more concise.

  • It would be better to add a table showing the clinical characteristics of patients

Unfortunately, we do not have a complete clinical record, but we have added a table (Table 1) with the available clinical data of the patients

  • The authors described NET G3 included NEC G3 (lines 146­­­–147), but NET and NEC are different disease concepts. Clinical data and OS for NET G3 and NEC G3 need to be evaluated separately.

In our sample, there were only 6 neuroendocrine carcinomas (NEC G3). Therefore, they were not used for statistical analysis.

  • Although the authors described “whereas mitotic count and semiquant Ki-67 were significantly different, the traditional evaluation of Ki-67 was not”, in Figure 3, traditional Ki-67 is marked with an asterisk and N. mitoses in not. Which is correct?

We have corrected Figure 3 which now having added other figures has become Figure 4.

  • I couldn't understand what the authors mean by table 2. Is this necessary information?

Table 2 gives correlation between the three measures of cellular proliferation and a couple of important clinical data. We used a table to summarize the data (all not significant) to make the text more readable avoiding to report all those not significant results.

  • In figure 4, the cumulative survival rates reach. Is this correct results? Please add the points of censored or events and number of patients at risk in the survival curves

We added the requested information in Figure 5 (previously fig. 4).  

  • There are a lot of evidence that the traditional method is significantly related to prognosis. Please discuss why the traditional method did not correlate with OS of GEP-NET patients in this study.

The current method for Ki-67 evaluation method is clearly very useful. We changed the Discussion to better explain that the new method we are proposing has several advantages in tumor classification, but it is not meant to completely replace the traditional evaluation. Rather, this paper advocates the need to refine this method by collecting more data. 

Round 2

Reviewer 2 Report

The manuscript has been revised well. However, there are a few nuclear points.

1) In Figure 5, the cumulative survival rates reach 0%. The prognosis for GEP-NET patients should not be that bad. Please check the data to make sure it is correct.

2) I could not understand that “t86 (lines 201, 215-217)” and “x24 (line 225)” mean. For readers who are not familiar with statistics such as me, there may be insufficient explanation.

3) Is "Figure 4" in line 20-22 a mistake for "Figure 5"?

Author Response

  • In Figure 5, the cumulative survival rates reach 0%. The prognosis for GEP-NET patients should not be that bad. Please check the data to make sure it is correct.

We checked the data and verified that it is correct. The figure is referring to differences in future outcomes when the survival is forced at zero for both conditions, to highlight the different trajectory of the curves for low / high.

  • I could not understand that “t86(lines 201, 215-217)” and “x2(line 225)” mean. For readers who are not familiar with statistics such as me, there may be insufficient explanation.

The numbers in subscript font after the name of the test (i.e., t86) represent the degree of freedom for the analysis. It’s a standard format to report statistical tests in most publications.

  • Is "Figure 4" in line 20-22 a mistake for "Figure 5"?

We corrected the name of the figure in the Results section.

Round 3

Reviewer 2 Report

The authors have correctly answered.